# Establishment of a Real-Time Reverse Transcription Recombinase-Aided Isothermal Amplification (qRT-RAA) Assay for the Rapid Detection of Bovine Respiratory Syncytial Virus

**DOI:** 10.3390/vetsci11120589

**Published:** 2024-11-24

**Authors:** Guanxin Hou, Siping Zhu, Hong Li, Chihuan Li, Xiaochen Liu, Chao Ren, Xintong Zhu, Qiumei Shi, Zhiqiang Zhang

**Affiliations:** Hebei Key Laboratory of Preventive Veterinary Medicine, Hebei Normal University of Science & Technology, Qinhuangdao 066600, China; hgx3430858@163.com (G.H.); 13846307826@163.com (S.Z.); hongli0711@163.com (H.L.); chihuan.li@outlook.com (C.L.); lxc1094576399@163.com (X.L.); 18832403751@163.com (C.R.); 14794363171@163.com (X.Z.); shiqiumei@126.com (Q.S.)

**Keywords:** bovine respiratory syncytial virus, qRT-RAA, specificity, sensitivity, detection assay

## Abstract

BRSV is one of the most important pathogens responsible for bovine respiratory syndrome and poses a threat to cattle production. Although high seroprevalence rates have been reported, the detection of the virus is much lower. The virus detection is often synchronized with the onset of respiratory symptoms, making the etiological testing of BRSV even more important for the prevention and control of the disease. In this study, we developed a qRT-RAA diagnostic method targeting the BRSV *F* gene based on emerging RAA technology, which allows the assay to be completed in 20 min at a constant temperature. This method is efficient and rapid, has high sensitivity and specificity, and is suitable for the clinical diagnosis of BRSV infection.

## 1. Background

Bovine respiratory syncytial virus (BRSV) is an important causative agent of bovine respiratory disease syndromes [1]. It has been widespread worldwide since its first identification in Europe in the 1970s. Presently, reports of BRSV isolation from cattle have been documented in Europe, the Americas, Africa, and Asia. BRSV infection mainly appears in newly weaned calves and young cattle in intensive farming, and the morbidity rate in calves can be as high as 60%, with a mortality rate of about 20%, causing significant losses [2]. In 2007, the first BRSV isolation in Heilongjiang province was reported in China [3]. BRSV infection is prevalent in cattle and cows, with a seropositivity rate of 41.2–94.4% and a virus detection rate of 0.95–5.3% [4,5], posing a significant threat to the bovine industry. BRSV mainly replicates in bronchiolar epithelial cells and type II pneumocytes, causing bronchial interstitial pneumonia, necrosis of epithelial cells, exudative or proliferative pneumonitis, and the appearance of syncytial cells, which was associated with replicating BRSV [6].

BRSV is an enveloped, non-segmented, negative-stranded RNA virus belonging to the genus Pneumovirus, subfamily Pneumovirus, and family Paramyxovirida [7]. The genome contains 10 open reading frames for encoding 11 proteins [8]. The F protein is located at the surface of the viral particle and is critical for viral replication by mediating the fusion of the viral envelope with the host cell membrane and the formation of syncytia. This protein contains 574 amino acids and is highly conserved among different BRSV isolates, frequently used as the target for detection methods design [9].

At present, enzyme-linked immunosorbent assay (ELISA), LAMP assay, and reverse transcription PCR (RT-PCR) are the major diagnostic methods for BRSV detection [10,11], requiring specialized instruments and personnel. Recombinase-aided isothermal amplification (RAA) is a method for the rapid amplification of nucleic acids at a constant temperature. The core of RAA technology is its special recombinase enzyme obtained from bacteria or fungi; it is able to attach tightly to the primer DNA at a low temperature and guide it to bind to the single DNA template formed by the single-stranded DNA-binding protein and complete the replication process under the action of DNA polymerase. Compared with conventional PCR technology, the whole amplification process is more efficient and rapid and escapes from the limitations of PCR instruments. Real-time monitoring of the RAA reaction process can be realized through the introduction of probes and the analysis of fluorescence signals [12,13,14]. For the characteristics of rapidity, simplicity, and high specificity and sensitivity, the real-time RAA is more favorable than other detection methods in actual pathogen detection. Here, we established a real-time reverse transcription RAA(qRT-RAA) method for BRSV by targeting the *F* gene and performed a comprehensive evaluation of its sensitivity, specificity, and effectiveness for clinical sample detection.

## 2. Materials and Methods

### 2.1. Virus and Clinical Samples

Viral nucleic acids of BRSV, IBRV, BPIV3, BVDV, BCoV were kept in our laboratory. Ninety-seven bovine samples (nasal swabs, oropharyngeal swabs) were collected by the Hebei Center for Prevention and Control of Animal Diseases and Control Center and stored at −80 °C.

### 2.2. Main Reagents and Materials

The pET-28a plasmid was kept in the laboratory, and Trans5α receptor cells were kept in the laboratory. Primers and probes were synthesized by Shanghai Sangong Bioengineering Co., Ltd. The Viral Nucleic Acid Extraction Kit was bought from Xian Tianlong Science Technology Co., Ltd. (Xian, China). The RT-RAA Nucleic Acid Amplification Kit (basic), The qRT-RAA Nucleic Acid Amplification Kit, and fluorescent probes were bought from Hangzhou Zongji Bio-Technology Co., Ltd. (Hangzhou, China). The T7 Ribo MAX Express Large Scale RNA Production System was purchased from Promega (Beijing, China), and the RNeasy Mini Kit was purchased from QIAGEN (Shanghai, China).

### 2.3. Design and Screening of Primers and Probes

The BRSV *F* gene sequence was downloaded from NCBI, and multiple sequence comparisons were performed using DNAMAN to select the highly conserved region. Three primer pairs were designed using Primer Premier 5.0 software (Table 1). The primers were preliminarily evaluated by the basic RT-RAA kit with testing for good specificity. The optimal primer pairs were selected. The probes were designed according to the optimal primer pairs. The fluorescence reporter group FAM and the fluorescence quenching group BHQ were labeled at the 30th base at the 5′ ends and the 17th base at the 3′ ends of the optimal probes, respectively, and the spacer (tetrahydrofuran, THF) was labeled in between the two groups.

### 2.4. Optimization of RT-RAA Reaction Conditions and System

The RT-RAA assay was developed using commercialized kits purchased from Hangzhou ZC Bio-Sci&Tech Co. Ltd. (Hangzhou, China). The reaction system was prepared according to the instructions of the commercialized kit and performed under different temperatures (37 °C, 38 °C, 39 °C, and 40 °C) for 30 min with 30 s per cycle. The amplification products were purified using the extraction solution. The purified product was electrophoresed and placed under a UV light for observation. The purified product was then reacted at 39 °C for different times (10 min, 20 min, 30 min, 40 min). Follow-up steps are the same as above.

### 2.5. Standard Plasmid for qRT-RAA

The genome of the BRSV sample was collected from 200 μL of the sample using the Viral Nucleic Acid Kit purchased by Tianlong Technology Co., Ltd. (Xian, China), which was utilized as the template. Partial sequence of the BRSV *F* gene was synthesized by one-step PCR. Recombinant plasmid pBRSV-F was constructed by cloning the PCR product into the pET-28a vector, which was sequenced and identified by Sangong Biotech (Shanghai, China) Co., Ltd.

### 2.6. Synthesizing RNA Standards by In Vitro Transcription

The pBRSV-F vector was extracted using a TaKaRa plasmid miniprep kit (Takara, Kusatsu, Japan), which was double enzymatic cleavage with restriction endonucleases (Sca I and Bam HI), after recovering and purifying the product, which was transcribed in vitro using Promega’s RiboMAXTM Express Large Scale RNA-T7 kit. Then, the product was purified by the RNeasy Mini Kit from QIAGEN, as described previously [15]. The concentrations of the cleaned RNA were measured using a spectrophotometer and passed through the formula: copies/μL = 6.02 × 10^23^ × mass concentration (ng/μL) × 10^−9^/(number of RNA bases × 340). The number of copies per microliter of the in vitro transcribed RNA standards was calculated, which was adjusted to 10^10^ copies/μL, which was diluted in a 10-fold serial dilution to achieve RNA standards ranging from 10^8^ to 10^0^ copies/μL.

### 2.7. qRT-RAA Reactions

As described previously [16], the reaction system was prepared according to the instructions of the commercialized kit. The system was mixed well before the reaction, with a setting for 20 min at 39 °C with 30 s per cycle, then spun briefly and transferred to the Genchek Fluorescence Detector (Beijing Musical Spectrum Diagnostic Technology Co., Ltd., Beijing, China)—real-time monitoring of the fluorescent signals.

### 2.8. Specificity Test

To evaluate the qRT-RAA specificity, the viral nuclear acids of BRSV, IBRV, BVDV, BPIV3, and BCoV were extracted using a commercial Virus DNA/RNA Kit (Tianlong Science and technology Co., Ltd.) and taken as amplification samples. Nucleic acid-free water was taken as a negative control. We performed the qRT-RAA reaction at 39 °C for 20 min with 30 s per cycle. The above experiments were carried out using the Genchek Fluorescence Detector. The experiment was repeated three times for all samples, and similar results were obtained.

### 2.9. Sensitivity Test

Nucleic acid-free water was used as a negative control. At the same time, the conventional PCR method [17] was used to compare and analyze the results of the two methods to evaluate the sensitivity of the qRT-RAA method developed in this research. This experiment was repeated at least three times, and similar results were obtained.

### 2.10. Repeatability Tests

Three concentrations of 10^8^ copies/μL, 10^6^ copies/μL, and 10^4^ copies/μL were selected as templates, and three repetitions were made for the intra-batch reproducibility test; at the same time, the above concentrations of standards were taken, and three repetitions of the qRT-RAA assay were carried out under the same conditions at different times as the inter-batch reproducibility test. To assess the reproducibility of the qRT-RAA method developed in this research. We repeated the experiment three times.

### 2.11. Detection of Clinical Samples

The viral nucleic acids of the ninety-seven bovine tissue samples were extracted using the Viral Nucleic Acid Kit purchased by Tianlong Technology Co., Ltd. (Xi’an, China), which was used as the template using the qRT-RAA method built up in this experiment, and the results of the assay were compared with the conventional PCR method [13] to compare the results of the two assays.

## 3. Results

### 3.1. Screening Primers

Three primer pairs, BRSVF1/R1, BRSVF2/R2, and BRSVF3/R3, were designed and evaluated by an agarose gel electrophoresis-based kit of ZC Bioscience™ Essential (Hangzhou, China) to obtain the optimal primers for qRT-RAA assays. In comparison, BRSVF1/R1 had the highest amplification efficiency (Figure 1A). The primers were followed by evaluating for specificity and showed high specificity without cross-reactivity with Infectious Bovine Rhinotracheitis Virus (IBRV), Bovine Parainfluenza Virus Type 3 (BPIV3), Bovine Viral Diarrhea Virus (BVDV), and Bovine Coronavirus (BCoV) (Figure 1B). Therefore, BRSV F1/R1 was used in the following study.

### 3.2. Optimization of Reaction System

Temperature and time are critical factors for the RAA assay. For the optimization of optimal temperature and reaction time, a reverse transcription RAA (RT-RAA) was performed for the initial screening. RT-RAA assay was performed under a range of temperatures (37 °C, 38 °C, 39 °C, and 40 °C), and we found that the method allows for working at a temperature ranging from 37 °C to 40 °C, with an optimal temperature of 39 °C (Figure 1C). After that, the reactions were performed at 39 °C for 10, 20, 30, and 40 min to determine the optimal amplification time. And 20 min was picked for excellent performance (Figure 1D). Therefore, a 39 °C incubation temperature and 20 min reaction time were used in the following experiments.

### 3.3. Specificity Analysis of qRT-RAA

Based on the optimal primers screened, a fluorescently labeled probe was designed to generate the qRT-RAA method, and as shown in Figure 2A, the developed method displayed a strong, productive amplification with the RNA standards of BRSV as a template. The specificity of qRT-RAA was assayed using nucleic acid templates of BRSV and other bovine viruses, such as IBRV, BVDV, BCoV, and BPIV3. The data showed that this method exhibited a strong fluorescent signal only when detecting BRSV nucleic acid, and no cross-reaction with other bovine viruses was observed, indicating a high specificity of this qRT-RAA method.

### 3.4. Sensitivity Analysis of qRT-RAA

RNA standards were serially 10-fold diluted and taken as templates to determine the sensitivity of the qRT-RAA method. The RNA standard was serially diluted to reach a concentration range of 10^8^ to 10^0^ copies/μL. As shown in Figure 2C, the RT-RAA method could detect as low as 10^2^ copies/μL of target RNA, while RT-RAA displays a detection limit of 10^4^ copies/μL (Figure 2D). Conventional PCR assays ranged from 10^8^ to 10^4^ copies (Figure 2D), suggesting a higher sensitivity of the RT-RAA assay.

### 3.5. Repeatability Analysis of qRT-RAA

Three concentrations of RNA standards at 10^8^ copies/μL, 10^6^ copies/μL, and 10^4^ copies/μL were used as templates in triplicate. After conducting the qRT-RAA assay developed in this experiment, inter-batch and intra-batch reproducibility tests were carried out. The results showed that the differences in the peak starting time of the same concentration of RNA standard were small in both intra-batch tests (Figure 3A) and inter-batch tests (Figure 3B), indicating that the qRT-RAA assay for BRSV developed in this research has good reproducibility.

### 3.6. Evaluation of Samples Using RT-RAA and Conventional PCR

Ninety-seven samples (nasal swabs, oropharyngeal swabs) were collected and tested by qRT-RAA and conventional RT-PCR assays, respectively. The data (Table 2) showed that seven samples were detected positive by both the qRT-RAA method and RT-PCR assay, with a positivity rate of 7.2% (7/97).

## 4. Discussion

BRSV is the major causative agent of respiratory diseases in dairy and beef cattle farming, which dramatically increases the mortality rate and cost of medication, causing significant losses in the cattle industry [18]. As commercial vaccines are hardly available, the present measures for BRSV control mainly rely on timely symptomatic treatment based on accurate diagnosis [19]. Although BRSV is capable of replicating in a variety of bovine-derived cells, this virus is difficult to isolate from clinical samples because of its fragile viral particles, poor environmental tolerance, and poor cellular adaptation, making it challenging for BRSV diagnosis [20]. Consequently, the establishment of a rapid, simple, and reliable BRSV detection method to eliminate positive cattle is particularly important to reduce the losses caused by this disease.

The *F* gene of BRSV is highly conserved with 97% to 99% amino acid homology among different isolates and is frequently used as a target gene for the design of nucleic acid assays [21]. In this study, we also chose this gene as the target gene to design the longer primers and probes used for generation of qRT-RAA method. To improve the efficiency of primer screening, we initially screened the primers with an RT-RAA kit by electrophoresis analysis. After determining the optimal primers, we designed probes within the amplification region and established and optimized the qRT-RAA method. The advantage of qRT-RAA technology is the ability to accomplish nucleic acid amplification efficiently under constant temperature within a short time, which relies on the involvement of specific recombinases, single-stranded proteins, etc. The qRT-RAA method established in this study for BRSV detection allows completion of the reaction and obtained results within 20 min at a constant temperature of 39 °C, which is more convenient and faster than the established RT-PCR, fluorescence timed PCR [22], Nano-PCR [23], ELISA, and immunofluorescent antibody assays [24]. Furthermore, the shortened reaction time did not diminish the sensitivity of the qRT-RAA assay, and the minimum detection limit of the qRT-RAA assay developed in this study was 102 copies/μL, basically the same as that of the Nano-PCR and real-time reverse transcription PCR (qRT-PCR) previously reported, but 100-fold higher than that of the conventional RT-PCR. The LAMP (loop-mediated isothermal amplification) [25] method is another effective way of achieving constant temperature amplification, and this approach involves the synergy of multiple primer pairs, requiring high requirements for primer design. In contrast, the primer design of the qRT-RAA method is the same as that of the conventional real-time reverse transcription PCR (qRT-PCR) method, with only slight modifications in primer length and probe labeling positions. Moreover, the developed qRT-RAA assay showed high specificity and did not cross-react with other common respiratory viruses, such as IBRV, BVDV, BPIV3, BCoV, etc.

Constant temperature and short reaction time are the advantages of the RT-RAA method, while the extraction of viral RNA did take more time. For the interference of proteins and the instability of viral RNA, crude samples are unsuitable for use as RT-RAA templates, and we will further explore the quick method for RT-RAA template preparation.

Although the high serologic positivity rate for BRSV has been reported in many articles, the detection rate of BRSV virus is much lower, ranging from 0.84% to 32.75% [26], indicating a long antibody persistence period post-infection and the necessity of etiologic testing. In this study, we collected 97 clinical samples from Hebei Province, China, for BRSV detection by the generated qRT-RAA, and seven were determined to be BRSV-positive; the positive rate was 7.2%. We noticed a perfect agreement of results from the generated qRT-RAA method in the present study and a conventional RT-PCR method previously reported. This may be due to the limited sample size and low BRSV detection rate since there were only seven positive samples. In addition, the samples we collected were all from cattle with significant respiratory symptoms, usually suggesting severe infection and high viral load. This may also have contributed to this agreement, which is consistent with the testing results used.

In summary, we have developed a qRT-RAA method for the rapid, sensitive, and specific detection of BRSV.

## 5. Conclusions

In conclusion, we established a qRT-RAA method for the detection of BRSV. This method displays high specificity and sensitivity and can be completed within 20 min at 39 °C, which can be applied for clinical testing of BRSV.

## Figures and Tables

**Figure 1 vetsci-11-00589-f001:**
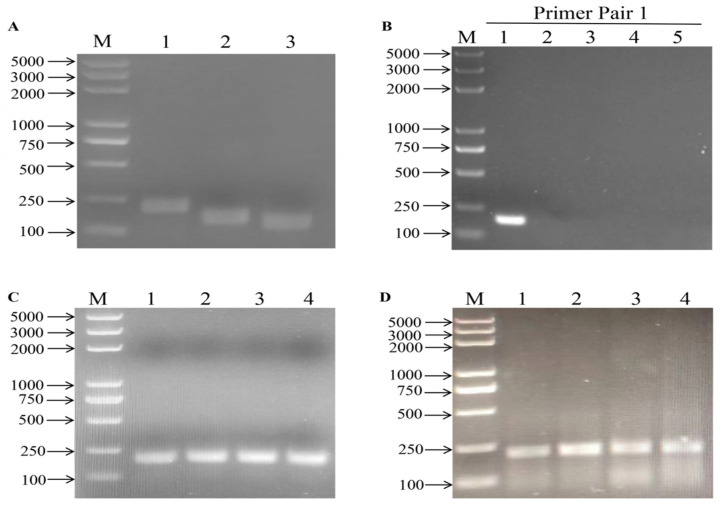
Determination of optimal primers and reaction conditions for BRSV RT-RAA methods. (**A**) Lane M: DL2000 DNA Marker (Takara). Determination of optimal primer pairs for BRSV RT-RAA that target the *F* gene (numbers 1–3 correspond to BRSVF1/R1, BRSVF2/R2, and BRSVF3/R3). A 5 μL template containing 10^4^ copies/μL of *F* gene RNA was added to each reaction system and carried out at 39 °C for 30 min. The amplification product was analyzed by electrophoretic in 1% (*w*/*v*) agarose gel. (**B**) Initial specificity assays. The primer pairs BRSV F1/R1 and 5 μL template were added (numbers 1–5 correspond to BRSV, BVDV, BPIV, IBRV, and BCoV). The reactions were performed at 39 °C for 30 min. (**C**) Results of optimal temperature screening for BRSV RT-RAA. Primer pair BRSV F1/R1 was selected for amplification using 5 μL template harboring 10^4^ copies/μL of *F* gene RNA with an incubation time of 30 min and different incubation temperatures (numbers 1–4 correspond to 37 °C, 38 °C, 39 °C, and 40 °C). (**D**) Results of optimal time screening for BRSV real-time fluorescent RT-RAA. Primer pair BRSV F1/R1 was selected for basal-type amplification using *F* gene RNA as a template. The incubation temperature was 39 °C, and the incubation time was different (numbers 1–4 correspond to 10 min, 20 min, 30 min, and 40 min). All the amplification products were electrophoresed on 1% agarose gel.

**Figure 2 vetsci-11-00589-f002:**
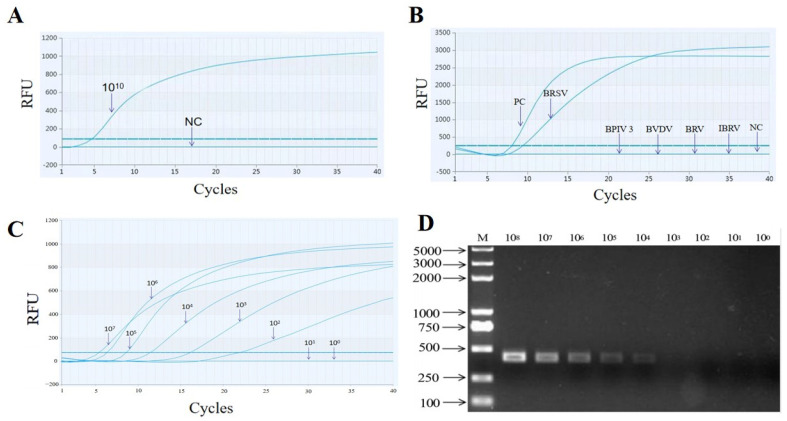
Evaluation of the BRSV RT-RAA detection method. RFU, Relative Fluorescence Unit. Each cycle denotes a 30 s reaction time. (**A**) The RT-RAA was performed with an RNA standard of 10^10^ copies under 39 °C for 20 min. (**B**) Specificity analysis of RT-RAA assays. Nucleic acids of BRSV, BVDV, BPIV, IBRV, BCoV, and ddH_2_O were extracted and used as templates for RT-RAA assay under optimal conditions. (**C**) Sensitivity evaluation. RNA standards were serially 10-fold diluted to a range of 10^8^ to 10^0^ copies/μL and subjected to RT-RAA assays. (**D**) M: DL2000 DNA Marker (Takara). The former diluted RNA standards were taken as templates for typical PCR, as reported previously.

**Figure 3 vetsci-11-00589-f003:**
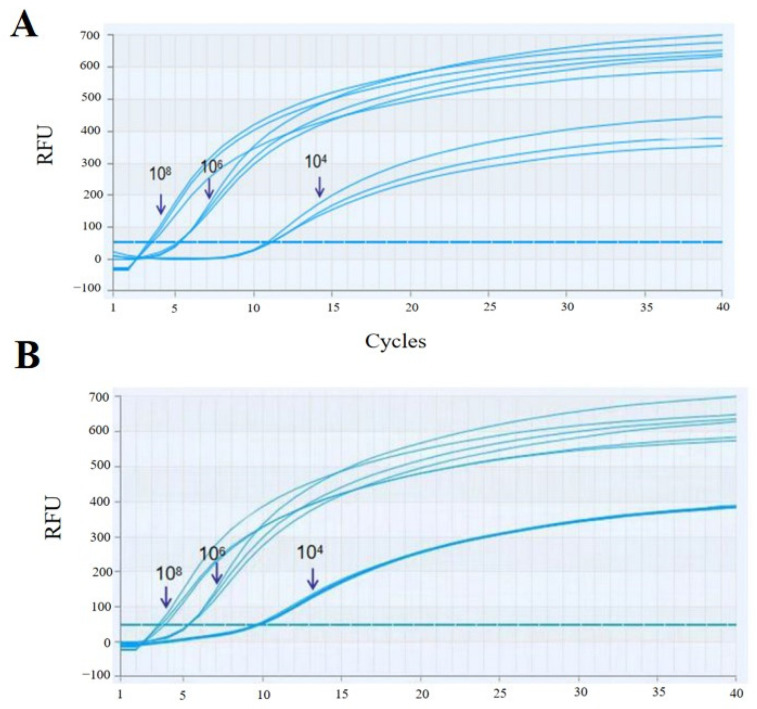
Repeatability tests. RNA standards with concentrations of 10^8^, 10^6^, and 10^4^ copies/μL were assayed using the RT-RAA system at 39 °C for 20 min. Each cycle denotes a 30 s reaction time. RFU is Relative Fluorescence Unit. (**A**) Inter-batch repeat test; (**B**) intra-batch repeat test.

**Table 1 vetsci-11-00589-t001:** Primer sequences screened in the RT-RAA assay.

Primer	Nucleotide Sequence (5′-3′)	Amplification Product Size (bp)
BRSV F1	TGTCAAGTAATGTTCAAATAGTYAGGCAAC	208 bp
BRSV R1	CAATACCACCCACGATCTGTCCTAGTTAAG	
BRSV F2	ACTAGCAAAGTACTCGATCTAAAGAACTAT	154 bp
BRSV R2	TTACACTAAATTCTCTAGCAATTTCTAACA	
BRSV F3	TGGTATTACCACACCCCTTAGTACATACATG	131 bp
BRSV R3	TCTGTTGCCTGACTATTTGAACATTACTTGA	

**Table 2 vetsci-11-00589-t002:** Evaluation of clinical samples by qRT-RAA and RT-PCR.

Assay		RT-PCR	
		Positive	Negative
qRT-RAA	Positive	7	0
	Negative	0	90

## Data Availability

The raw data supporting the conclusions of this article will be made available by the authors without undue reservation.

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
