# Peer review of "Establishment of a Real-Time Reverse Transcription Recombinase-Aided Isothermal Amplification (qRT-RAA) Assay for the Rapid Detection of Bovine Respiratory Syncytial Virus"

_vetsci, 2024, doi:10.3390/vetsci11120589_

Round 1
Reviewer 1 Report
Comments and Suggestions for Authors
This manuscript by Hou described establishment of a real-time RT-RAA assay for the detection of BRSV. This method would be useful for survey of BRSV in the field. This manuscript only described a methodical establishment, in my opinion, it is too simple to be accepted. In the meantime, there is a total of three figures in this manuscript. Unfortunately, Fig2 and Fig3 wereoverlapped together. Thus, I could not evaluate those results.
Author Response
Comments1:
This manuscript by Hou described establishment of a real-time RT-RAA assay for the detection of BRSV. This method would be useful for survey of BRSV in the field. This manuscript only described a methodical establishment, in my opinion, it is too simple to be accepted. In the meantime, there is a total of three figures in this manuscript. Unfortunately, Fig2 and Fig3 were overlapped together. Thus, I could not evaluate those results.
Response1:
Thank you for the comment, We are sorry for the confusion caused by the figures and have made corrections in the revised version of manuscript.
Reviewer 2 Report
Comments and Suggestions for Authors
Summarize
The authors have developed and optimized detection method using RT-RAA for rapid detection of BRSV, which are the main cause of bovine respiratory diseases. They designed primers for the F gene, which has a highly conserved sequence, optimized the assay, and verified its specificity and sensitivity. As a result, this method showed high specificity and reproducibility and was 100 times more sensitive than the conventional method. In addition, when validation was performed using bovine clinical samples, the test demonstrated 100% sensitivity and specificity compared to conventional RT-PCR methods. The authors conclude that this RT-RAA method is useful for rapid and accurate detection of BRSV in clinical settings.
Major comments
The results are clear and the arguments are easy to understand. However, I feel that the overall structure of the text and some points need supplementation and correction.
Revisions (V2 and 3) have broken the style and make review difficult. Generate a PDF with simple changelog and review it before submission. In particular, Figure 2 cannot be seen because Figure 3 overlaps, so I refer to version 1 for review.
Please check, correct or comment on the following points:
1. In lines 12, 43 and 45, There are different notations for RAA and RT-RAA, and the full form should be defined accurately and uniformly.
2. The mechanism of RT-RAA and the characteristics of its primers, which differ from other techniques, should be explained in more detail in the introduction rather than in the discussion section.
3. It is necessary to clearly state whether it is the normal RT-RAA method (electrophoresis) or the RT-RAA method using real-time detection. This is sometimes written incorrectly and is difficult to understand.
4. In Figure 1, the amount of each template added should be specified.
5. In Figures 2 and 3, I think that "Cycles" in the diagram for real-time detection of RT-RAA is one cycle in 30 seconds, but this should be clearly stated or the time notation should be changed. Also, the full form of RFU (relative fluorescence unit ?) should be clearly stated in the figure or legend.
6. In Figure 2B, All viruses other than IBRV are RNA viruses, so shouldn't they use RNA as a template?
7. In Figure 2C, Why is 105 copies/µL applied sample so far off the expected position? If it happens repeatedly, please discuss it.
8. In lines 99, 103 and 106(in Section 2.4), Is it a mistake to write "copies" as "copies/µL"? Furthermore, is the PCR lower detection limit of 5x102 a mistake for 102?
9. In Table 2, This is a very strange. If the results of ordinary PCR (nested RT-PCR ?) and RT-RAA are in perfect agreement, the table should look like the one shown below. If not, please explain. Also, the number of positive samples is too small to evaluate the sensitivity.
Ordinary
positive negative
RT-RAA positive 3 0
RT-RAA negative 0 27
10. In Table 2, is the reason why the results do not change even if the sensitivity is increased 100 times because the virus titer at the time of infection is high? If so, there is no point in increasing the sensitivity. Please add your opinion to the discussion section.
11. In Discussion, If RNA purification is required in clinical practice, it would be less quick, convenient, and economical. Is it possible to use crude samples?
12. In Authors’ contribution, please make sure that the contributions of the authors are legitimate in accordance with the authorship requirements. Normally, only conducting a literature search or knowledge summarization does not qualify as an author.
Minor comment
Please correct the following points:
1. viruses other than BRSV and LAMP should be stated in full form when they are first mentioned in the main text.
2. Please provide details (swab…) about the 40 bovine-derived clinical samples not only in the main text but also in the method Clinical sample section.
3. In Abbreviation section, RT-PCR is reverse-transcription PCR used in section 2.6, not a real-time PCR.
Comments on the Quality of English LanguageThere are also many English spelling mistakes, missing words, and spacing mistakes. Please check again carefully.
Author Response
Thank you for the comments concerning our manuscript. These comments are all valuable and very helpful for revising and improving our paper. We have studied comments carefully and have made correction which we hope meet with approval. Revised portion are marked with red colors in the paper. The main corrections in the paper and the responds to the reviewer's comments are as following:
Comments 1: In lines 12, 43 and 45, There are different notations for RAA and RT-RAA, and the full form should be defined accurately and uniformly.
Response 1: Thank you for your comment, we are sorry for the misleading statement and have made corrections in the text.
Comments 2:The mechanism of RT-RAA and the characteristics of its primers, which differ from other techniques, should be explained in more detail in the introduction rather than in the discussion section.
Response 2: Thank you for the comment. We've made modification as required.Please see lines 54-60.
Comments 3: It is necessary to clearly state whether it is the normal RT-RAA method (electrophoresis) or the RT-RAA method using real-time detection. This is sometimes written incorrectly and is difficult to understand.
Response 3: Thank you for your comment, we have made corrections in the article. RT-RAA is used to define conventional reverse transcriptase RAA method(electrophoresis), whereas qRT-RAA is used to refer to real-time reverse transcriptase RAA method.
Comments 4: In Figure 1, the amount of each template added should be specified.
Response 4: We've made the corrections. Please see lines 93-98.
Comments 5: In Figures 2 and 3, I think that "Cycles" in the diagram for real-time detection of RT-RAA is one cycle in 30 seconds, but this should be clearly stated or the time notation should be changed. Also, the full form of RFU (relative fluorescence unit ?) should be clearly stated in the figure or legend.
Response 5: Thank you for the comment. We've made modification as required. Please see lines 135,168,242,262 and 269.
Comments 6: In Figure 2B, All viruses other than IBRV are RNA viruses, so shouldn't they use RNA as a template?
Response 6: In this study, we performed specificity analysis by using viral nucleic acids extracted with commercial kits as templates. To this, we have renewed the description. Please see lines 266-267.
Comments 7: In Figure 2C, Why is 105 copies/µL applied sample so far off the expected position? If it happens repeatedly, please discuss it.
Response 7: We apologize for the misleading picture, it was an unrepeatable exceptional data, we have re-validated it and re-uploaded a new figure. Please see lines 116.
Comments 8: In lines 99, 103 and 106(in Section 2.4), Is it a mistake to write "copies" as "copies/µL"? Furthermore, is the PCR lower detection limit of 5x102 a mistake for 102?
Response 8: We are sorry for the mistakes an have made the corrections. Please see lines 113-114.
Comments 9: In Table 2, This is a very strange. If the results of ordinary PCR (nested RT-PCR ?) and RT-RAA are in perfect agreement, the table should look like the one shown below. If not, please explain. Also, the number of positive samples is too small to evaluate the sensitivity.
Ordinary
positive negative
RT-RAA positive 3 0
RT-RAA negative 0 27
Pesponse 9: We agree and have made some modifications. We are sorry for the mistakes an have made the corrections. As for the positive sample size, we updated the data based on samples collected from the Hebei Province, China in 2024, however the number of positive samples is still small, and we have also described it in the discussion. Please see lines 209-212.
Comments 10: In Table 2, is the reason why the results do not change even if the sensitivity is increased 100 times because the virus titer at the time of infection is high? If so, there is no point in increasing the sensitivity. Please add your opinion to the discussion section.
Response 10: Thank you for the comment. We believe that the small number of positive samples is the main cause for this high concordance of qRT-RAA and ordianary RT-PCR. In addition, the samples we collected were all from cattle with significant respiratory symptoms, usually accompanied with severe infection and high viral load, may also contributed this agreement. To this we add the explanation in discussion section. Please see lines 209-212.
Comments 11: In Discussion, If RNA purification is required in clinical practice, it would be less quick, convenient, and economical. Is it possible to use crude samples?
Response 11: Thank you for the comment. The points you raised are very meaningful. Constant temperature and short reaction time are the advantages of RT-RAA method, while the extraction of viral RNA did take more time. For the interference of proteins and the instability of viral RNA, crude samples are unsuitable for using as RT-RAA templates and we will further explore the quick method for RT-RAA template preparation. We have also explained it in the discussion section. Please see lines 202-204.
Comments 12: In Authors’ contribution, please make sure that the contributions of the authors are legitimate in accordance with the authorship requirements. Normally, only conducting a literature search or knowledge summarization does not qualify as an author.
Response 12: Thank you for the comment. We have confirmed the authorships and made some modifications. Please see lines 310-313.
Comments 13: viruses other than BRSV and LAMP should be stated in full form when they are first mentioned in the main text.
Response 13: Thank you for the comment. We have added the information as required. Please see lines 70-72.
Comments 14: Please provide details (swab…) about the 40 bovine-derived clinical samples not only in the main text but also in the method Clinical sample section.
Response 14: Thank you for the comment. We have added the information as required. Please see lines 220-221.
Comments 15: In Abbreviation section, RT-PCR is reverse-transcription PCR used in section 2.6, not a real-time PCR.
Response 15: We've made the correction. Please see lines 290.
Comments 16: There are also many English spelling mistakes, missing words, and spacing mistakes. Please check again carefully.
Response 16: Thank you for the comment. We have reviewed the whole manuscript and made corrections.
Reviewer 3 Report
Comments and Suggestions for Authors
The Editor Veterinary Sciences
Thank you for the opportunity to review the manuscript: “Establishment of a real-time recombinase-aided isothermal amplification (RT-RAA) assay for the rapid detection of bovine respiratory syncytial virus”. The paper has been carefully reviewed but significant concerns arose:
Line 27: BRSV is an important pathogen; however, it cannot be considered the main one, but rather one of the most significant within cattle farming.
The incidence of this agent is not described at any time in the study area, justifying the implementation of a new technique.
Regarding the costs per exam, is there any advantage to the RT-RAA?
It is not possible to evaluate the cycle graphs in figure 2 as they are altered and not included in the supplementary material.
What equipment was used to keep the samples at a constant temperature?
Figure 3 is cited in the text but does not appear in the article.
In the discussion, vaccination is mentioned, but a question arises: can it interfere with this type of diagnosis?
Overall, the work presents good quality, having merits for its publication. Its deficiencies are easily remediable. I believe the cost issue of the new technique should be addressed. I think the results are still preliminary for a change in technique, considering that only three samples were effectively detected. RT-RAA should undergo further validation before being considered definitive for diagnosis, but as presented, it shows a good diagnostic alternative.
Author Response
Thank you for the comments concerning our manuscript. These comments are all valuable and very helpful for revising and improving our paper. We have studied comments carefully and have made correction which we hope meet with approval. Revised portion are marked with red colors in the paper. The main corrections in the paper and the responds to the reviewer's comments are as following:
Comments 1: Line 27: BRSV is an important pathogen; however, it cannot be considered the main one, but rather one of the most significant within cattle farming.
Response 1: Thank you for the professional review, you are right, and we have revised the description. Please see lines 19,38.
Comments 2: The incidence of this agent is not described at any time in the study area, justifying the implementation of a new technique.
Response 2: Thank you for the comment. We have already illustrated this in the main text.Please see lines 41-42.
Comments 3: Regarding the costs per exam, is there any advantage to the RT-RAA?
Response 3: You ask a very practical question, and the fact is that the RT-RAA method remains relatively expensive now, but it is decreasing recently, and we believe that it will be affordable enough for livestock production in the future with the maturity of the production technology.
Comments 4: It is not possible to evaluate the cycle graphs in figure 2 as they are altered and not included in the supplementary material.
Response 4: Thank you for the comment, we have added an explanation to cycles in the text. Please see lines 135,168,242,262 and 269.
Comments 5: What equipment was used to keep the samples at a constant temperature?
Response 5: The real-time fluorescence RAA method established in this paper needs to be performed in a real time PCR system or a dedicated thermostatic fluorescence monitoring instrument. As for the conventional electrophoresis-based RAA assay, it could be performed in any thermostatable devices, even a thermostatic water bath.
Comments 6: Figure 3 is cited in the text but does not appear in the article.
Respones 6: Thank you for the comment, We are sorry for the confusion caused by the figures and have made corrections in the revised version of manuscript. Please see lines 152.
Comments 7: In the discussion, vaccination is mentioned, but a question arises: can it interfere with this type of diagnosis?
Response 7: Thank you for the comment. Due to the low immunization rate of the attenuated BRSV vaccine and the unavailability of the vaccine in many countries and regions, we did not strive to avoid the interference of the vaccine virus in designing this diagnostic method.
Round 2
Reviewer 2 Report
Comments and Suggestions for Authors
The authors have generally responded honestly and appropriately to my comments, however one critical issue remains.
The authors are evaluating the qRT-RAA method using clinical samples, but there are several places where the number of clinical samples and the number of positives have not been corrected. Please carefully review the entire text and correct it.
As I pointed out last time, Table 2 has not been corrected and must be revised to match the main text.
Author Response
Thank you for the comments concerning our manuscript. These comments are all valuable and very helpful for revising and improving our paper. We have studied comments carefully and have made correction which we hope meet with approval. Revised portion are marked with red colors in the paper. The main corrections in the paper and the responds to the reviewer's comments are as following:
Comments 1: The authors are evaluating the qRT-RAA method using clinical samples, but there are several places where the number of clinical samples and the number of positives have not been corrected. Please carefully review the entire text and correct it.
Response 1: Thank you for the comment. We are sorry for the mistakes an have made the corrections. Please see lines 171-174, 220 and 282.
Comments 2: As I pointed out last time, Table 2 has not been corrected and must be revised to match the main text.
Response 2: Thank you for the comment. We are sorry for the mistakes and have made modifications to Table 2. Please see lines 174.